# Methanol-essential growth of *Escherichia coli*

Fabian Meyer[1], Philipp Keller[1], Johannes Hartl[1], Olivier G. Gröninger[1], Patrick Kiefer[1] & Julia A. Vorholt[1]

Methanol represents an attractive substrate for biotechnological applications. Utilization of reduced one-carbon compounds for growth is currently limited to methylotrophic organisms, and engineering synthetic methylotrophy remains a major challenge. Here we apply an in silico-guided multiple knockout approach to engineer a methanol-essential *Escherichia coli* strain, which contains the ribulose monophosphate cycle for methanol assimilation. Methanol conversion to biomass was stoichiometrically coupled to the metabolization of gluconate and the designed strain was subjected to laboratory evolution experiments. Evolved strains incorporate up to 24% methanol into core metabolites under a co-consumption regime and utilize methanol at rates comparable to natural methylotrophs. Genome sequencing reveals mutations in genes coding for glutathione-dependent formaldehyde oxidation (*frmA*), NAD(H) homeostasis/biosynthesis (*nadR*), phosphopentomutase (*deoB*), and gluconate metabolism (*gntR*). This study demonstrates a successful metabolic re-routing linked to a heterologous pathway to achieve methanol-dependent growth and represents a crucial step in generating a fully synthetic methylotrophic organism.

---

[1] Institute of Microbiology, Department of Biology, ETH Zurich, Zurich 8093, Switzerland. Correspondence and requests for materials should be addressed to J.A.V. (email: jvorholt@ethz.ch)

The conversion of one-carbon compounds is a major challenge, as it necessitates the introduction of C–C bonds, upon which all life hinges. While autotrophs, which are capable of carbon fixation from $CO_2$, require a separate energy source, methylotrophs utilize reduced one-carbon compounds to synthesize all cellular components and harvest energy for growth. Prominent and important reduced one-carbon compounds are methanol and methane. Methane is the most abundant fossil fuel compound on Earth, and it is 25 times more potent as a greenhouse gas than $CO_2$, and a key component in the carbon cycle[1]. In addition, methane becomes available through renewable biomass conversion processes. Methanol can be produced from methane[2–6] or many other sources such as syngas, a mixture of mainly $H_2$ and CO[7], from which it is already produced in megaton scales[8]. Methanol is thus a readily available substrate and led to the concept of a methanol-based economy[9].

The metabolism of natural methylotrophs, which allows them to convert methanol into biomass, has potential for methanol-based biotechnology. Work in the last 50 years has revealed several conserved pathways, including one-carbon dissimilation pathways consisting of nicotinamide adenine dinucleotide (NAD)- or pyrroloquinoline quinone (PQQ)-dependent methanol dehydrogenases, as well as formaldehyde oxidation pathways that depend on cofactors such as tetrahydromethanopterin or glutathione, and assimilation pathways, i.e., the ribulose monophosphate pathway, the serine cycle, and the Calvin cycle[10, 11]. There is increasing interest in utilizing natural methylotrophs, particularly for specialty chemical production[12–22]. Nonetheless, product spectra, genetic tools, and product yields need to be improved for this technology to become sustainable.

An alternative approach to exploit methanol in biotechnology is the engineering of synthetic methylotrophy[23–25]. Initial studies using *Escherichia coli*[26] and *Corynebacterium glutamicum*[27] as host organisms provided proof of concept. By introduction of a methanol dehydrogenase (Mdh) and the ribulose monophosphate pathway into the heterologous hosts, assimilation of methanol was demonstrated. Additional studies have since been carried out to further improve strain development and to modify additional hosts[28–31]. However, so far no organism capable of growing on methanol as the sole carbon and energy source has been engineered. Indeed, engineering a non-native carbon fixation cycle from reduced or oxidized one-carbon compounds in a heterotrophic host is a difficult task, and the mere presence of the synthetic pathway in the host does not automatically lead to the utilization of the introduced pathway[32].

In the present work, we report the generation of the first synthetic methanol-dependent strain in the presence of gluconate as co-substrate. We use theoretical considerations as a starting point to force the heterologous host, here *E. coli*, to utilize methanol as an essential substrate via a synthetic pathway and by coupling co-consumption of methanol to growth. This strategy allows us to harness the power of natural selection to optimize strain performance (Fig. 1a). Using isotopic steady-state labeling experiments, we demonstrate significant incorporation of methanol into core metabolites during exponential growth of the engineered and evolved strains (24% into hexose 6-phosphate) and show a yield increase of approximately 30% due to methanol

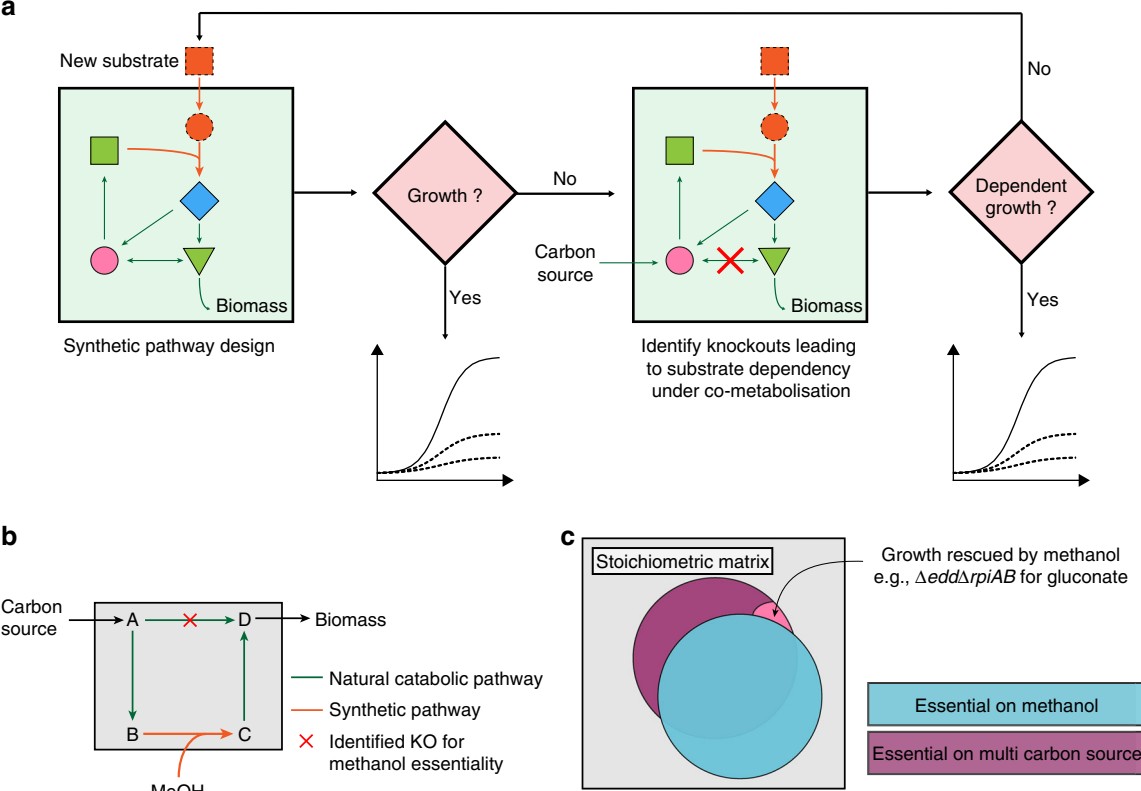

**Fig. 1** Strategy for exogenous substrate utilization. **a** Introduction of a synthetic pathway is validated in silico, tested for growth, and optimized using adaptive laboratory evolution. If growth is not possible, substrate utilization is coupled to cellular fitness under co-metabolization and optimized; otherwise, either pathway design is modified or new knockouts leading to substrate-dependent growth are tested. **b** Substrate (methanol) essential gene deletions are obtained by knockout of a natural catabolic pathway, thus allowing biomass formation only upon use of the synthetic methanol-dependent pathway. **c** Example of identified knockouts leading to methanol essentiality in the case of gluconate as a carbon source

assimilation, thus surpassing the lower stoichiometrically coupled methanol uptake.

## Results

**In silico identification of methanol-essential genotypes.** Although enzymes that allow *E. coli* to consume methanol have recently been successfully implemented, all strains engineered so far depend on multi-carbon substrates and grow without methanol addition[26, 28, 31, 33–35], which renders the transition to growth on methanol as sole carbon and energy source a major hurdle. In order to metabolically link growth to the consumption of methanol, we evaluated methanol essentiality in silico using flux balance analysis (Fig. 1b, c). We used the stoichiometric genome scale model of *E. coli* iAF1260[36] into which reactions catalyzed by Mdh, 3-hexulose-6-phosphate synthase (Hps), and 6-phospho-3-hexuloisomerase (Phi) were integrated. The model predicted growth as a function of methanol uptake. However, methanol as a sole carbon source is unable to support growth experimentally[26]. We thus developed a strategy to link methanol assimilation to growth in the presence of a supporting substrate using genetic constraints. To achieve this goal, we evaluated gene deletions leading to methanol essentiality and analyzed a set of multi-carbon sources as co-substrates in addition to methanol: acetate, gluconate, glucose, glycerol, pyruvate, ribose, xylose, and succinate. Reaction knockout (KO) analysis was performed using the open-source software FlexFlux[37] and the uptake rate for all carbon sources was set to 7 mmol gCDW$^{-1}$ h$^{-1}$ as an approximation (corresponding to glucose uptake rates under shake flask condition[38]). Applying an iterative workflow (Supplementary Fig. 1), we tested the different combinations of multi-carbon sources and methanol to allow for methanol-essential growth under mixed substrate conditions. Our goal was to identify gene deletions that resulted in failure to grow on a multi-carbon source alone, but resulted in growth when methanol was additionally present. Another constraint was that the potential for pure methylotrophic growth was retained (Fig. 1c). The minimum number of gene KOs leading to the desired coupling of a multi-carbon source and methanol was determined for the different carbon sources, a process for which gene redundancy and the high plasticity of the metabolic network represented a challenge. We applied an iterative workflow to explore multiple KO strategies until targets were identified. (i) In a first step, KO analyses of the multi-carbon source versus methanol were performed to identify KOs, which resulted in growth on methanol but led to no growth on the multi-carbon source alone (predicted growth rate ≤0.01 h$^{-1}$). KOs unable to be rescued by methanol such as transporters for the multi-carbon source were discarded for further analysis. (ii) For most multi-carbon sources, no metabolically feasible solution for methanol-essential growth based on single KOs was found, and growth rate comparisons of non-essential KOs was performed comparing the multi-carbon source versus co-consumption with methanol to identify highest differential fitness hits. (iii) The workflow was repeated with the identified gene targets until a genotype allowing methanol-dependent growth was identified. A metabolic solution for methanol-essential growth based on gene deletions for gluconate is indicated in Fig. 2 while other potentially feasible solutions based on the model iAF1260 are provided in the Supplementary Data 1. Performing the same analysis with the recently published updated model iML1515[39] gave similar results that are overall consistent with the ones shown in Fig. 2 (for details see Supplementary Data 2).

**In vivo validation of methanol-essential growth.** Based on our computational framework, we identified multiple solutions to couple methanol consumption to co-substrate-dependent growth. We subsequently focused on substrates leading to increased availability of ribulose-5-phosphate as a key intermediate required for methanol assimilation via Mdh and Hps. In consequence, we chose to experimentally test gluconate as a co-substrate as it leads to high pentose 5-phosphate pools[40].

Methanol-dependent growth in the presence of gluconate is predicted in silico by a KO of *edd* (encoding phosphogluconate dehydratase) in combination with *rpiAB* (encoding ribose-5-phosphate isomerases) in the presence of *mdh*, *hps*, and *phi* (Fig. 2). Gluconate is phosphorylated to 6-phosphogluconate, but its entry into the Entner–Doudoroff pathway is prevented by the *edd* KO, resulting in flux re-direction to ribulose 5-phosphate via decarboxylation catalyzed by 6-phosphogluconate dehydrogenase (Gnd). In addition, the accumulation of the formaldehyde acceptor ribulose 5-phosphate is fostered by a KO in *rpiAB*, blocking the interconversion of ribulose 5-phosphate to ribose 5-phosphate via the non-oxidative pentose phosphate pathway (Fig. 2). Together with methanol, growth can be rescued by condensation of formaldehyde with ribulose-5 phosphate to 3-hexulose 6-phosphate and subsequent isomerisation to fructose 6-phosphate.

Based on these findings, we generated a methanol-essential strain (MeSV1) by knocking out *edd* and *rpiAB* and introducing the plasmids pSEVA424 *mdh2* PB1 and pSEVA131 *hps phi* Mf, encoding Mdh from *Bacillus methanolicus* PB1 and Hps and Phi from *Methylobacillus flagellatus* (see Supplementary Table 2), which we had identified as suitable enzymes in a previous study[26]. Initial growth experiments of MeSV1 were performed using gluconate and methanol in the presence of small amounts of yeast extract (0.1 g L$^{-1}$), which resulted in a slight growth advantage due to methanol (<1 doubling; Supplementary Fig. 2a). To remove the initial metabolic burden, pyruvate was added to the culture since in silico analysis predicted a favorable flux directionality of the core metabolism, i.e., resembling methanol flux distribution. Using pyruvate and yeast extract as a helper substrate in addition to gluconate/methanol resulted in slow, but steady, growth of the strain while no growth was observed in the absence of methanol (Supplementary Fig. 2b). Methanol essentiality on gluconate in combination with pyruvate was not proposed on a stoichiometric level and might be due to metabolic regulation. Based on the consistent growth observed in the presence of methanol, an adaptive laboratory evolution (ALE) experiment was conducted in quintuplicates with the goal to improve co-consumption of methanol and gluconate, initially for five passages (about 35 generations), and in the presence of pyruvate. Transfer to medium containing only gluconate, methanol and small amounts of yeast extract (0.1 g L$^{-1}$) resulted in methanol-essential growth on gluconate (Fig. 3). The evolved cultures, termed methanol-essential strains version 1.1 (MeSV1.1), showed a threefold increase in final OD$_{600}$ compared to the engineered parental strain MeSV1 (Fig. 3, passages 8–12 compared to parental strain). Continued ALE did not improve these strains further, implying that a local optimum was reached. Although a significant growth increase of MeSV1.1 compared to the parental strain was achieved, full growth rescue with methanol (Fig. 3, dotted line) was not possible.

**Optimization of methanol-dependent growth.** The inability of MeSV1.1 to fully restore growth on gluconate with methanol and the apparent stagnant growth yield (measured by final OD$_{600}$) in the initial ALE experiments (Fig. 3) may have various underlying causes, including the accumulation of an intracellular toxic compound or dead-end product. Careful investigation of flux balance calculations revealed that most of the

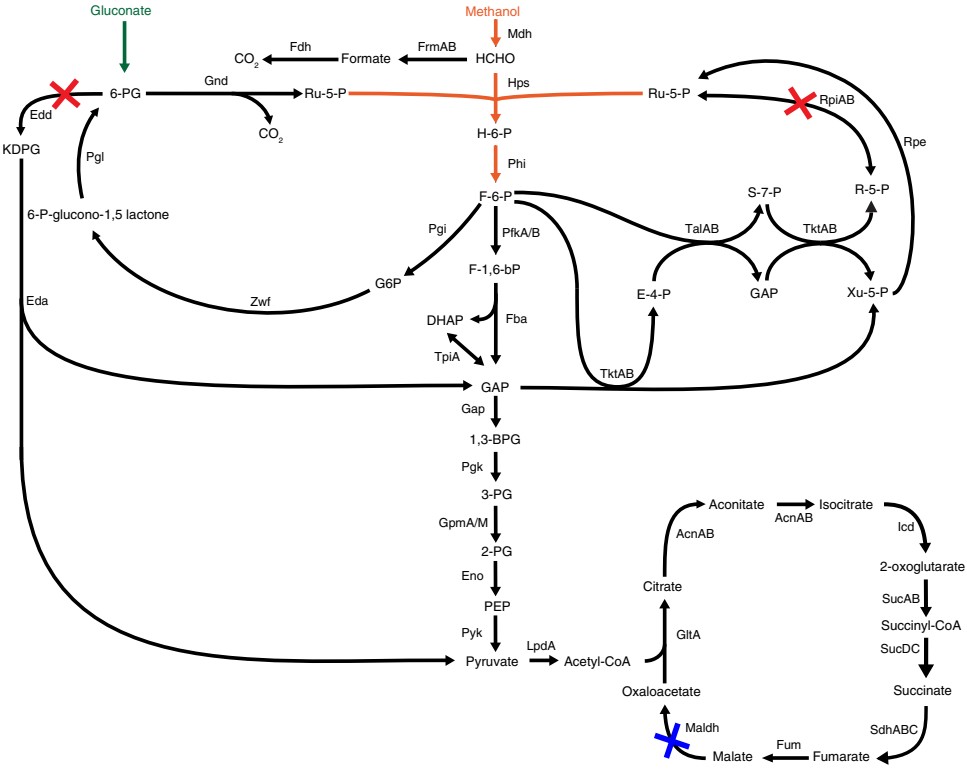

**Fig. 2** Metabolism of a methanol-essential *Escherichia coli* strain. The introduced synthetic pathway is depicted in orange and comprises methanol dehydrogenase (Mdh), 3-hexulose-6-phosphate synthase (Hps), and 6-phospho-3-hexuloisomerase (Phi). Identified knockouts leading to methanol essentiality in the case of gluconate as a carbon source are shown in red; phosphogluconate dehydratase (*edd*) and ribose-5-phosphate isomerase (*rpiAB*). The knockout for reducing TCA cycle activity and mimicking natural methylotrophs is shown in blue; here malate dehydrogenase (*maldh*). Names and abbreviations are given according to the Biocyc database (http://www.biocyc.org)

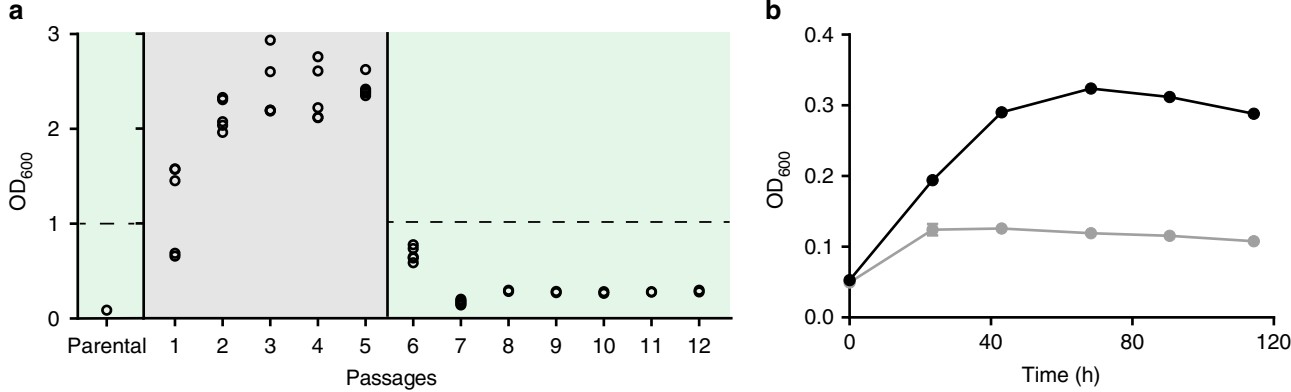

**Fig. 3** Evolution of methanol-essential strain version 1 (MeSV1). The parental strain used for adaptive laboratory evolution was $\Delta edd\Delta rpiA\Delta rpiB$ pSEVA424 *mdh2* PB1 pSEVA131 *hps phi* Mf. **a** Shown are maximal $OD_{600}$ values after each passage of MeSV1 ($n = 5$). Initially, hardly any growth was observed in the presence of 5 mM gluconate and 500 mM methanol. The addition of 20 mM pyruvate as a helper substrate initiated growth and was used to start the evolution experiments (gray). After five passages, growth without pyruvate was possible and resulted in a threefold higher final $OD_{600}$ compared to the parental strain (green). Control cultures without methanol did not grow beyond background levels due to 0.1 g L$^{-1}$ yeast extract ($OD_{600} \sim 0.13$). Final $OD_{600}$ of a wild-type *E. coli* strain containing empty plasmids under the same growth conditions reaches $OD_{600} = 1.06 \pm 0.02$ (dotted line). **b** Growth curve of one evolved replicate (MeSV1.1) after 12 passages ($n = 3$) with (black) and without (gray) methanol. Data presented as mean ± standard deviation

reduced nicotinamide adenine dinucleotide (NADH) available for energy generation is produced via the introduced Mdh, whereas a low tricarboxylic acid (TCA) cycle activity is predicted. On the contrary, on most multi-carbon sources including gluconate, a high TCA cycle activity is expected with which most reducing equivalents are produced. It is thus conceivable that a surplus

in NADH is generated upon mixed substrate conditions, which might distort redox balance and potentially result in growth arrest. From a thermodynamic viewpoint, accumulation of NADH would further negatively affect the initial step of methanol oxidation by increasing the Gibbs energy of the reaction. In this regard, it is interesting to note that obligate methylotrophs

lacking a closed TCA cycle exist[41, 42], and recent findings indicate that low TCA cycle activity is linked to methylotrophic growth in facultative methylotrophs[43, 44]. Indeed FBA confirmed that a closed oxidative cycle is not required for growth in the presence of gluconate and methanol.

Based on the above considerations, we set out to test the hypothesis of redox imbalance due to TCA activity and generated a KO of NAD-dependent malate dehydrogenase (maldh) in a $\Delta edd \Delta rpiAB$ genetic background, followed by introduction of

mdh, hps, and phi as above. The strain, termed MeSV2, was subjected to a first laboratory evolution experiment followed by five replicates (R1–R5) using the experimental setup described above. While the initial two passages showed decreased growth performance compared to MeSV1.1, the following passages led to major growth improvements (Fig. 4a). Additional passages without pyruvate, but still containing small amounts of yeast extract (0.1 g L$^{-1}$), resulted in improved growth on gluconate and methanol until the 16th passage. The obtained population MeSV2.1 and the replicate MeSV2.1 R cultures fully restored growth on gluconate with methanol and were even able to achieve slightly higher yields (OD$_{600}$ = 1.21 ± 0.01) compared to a wild-type E. coli under the same growth conditions (OD$_{600}$ = 1.06 ± 0.02) (Fig. 4a, c) with the exception of one replicate (MeSV2.1 R2). Control cultures cultivated without methanol did not grow beyond levels supported by the addition of 0.1 g L$^{-1}$ yeast extract alone.

Next, we optimized MeSV2.1 in the absence of yeast extract in a subsequent laboratory evolution experiment. Lack of yeast extract reduced the growth rate initially; however, upon nine passages, an increased optical density maximum (OD$_{600}$ = 1.34 ± 0.03) was achieved that was even higher than values obtained with yeast extract supplementation. The growth rate improved significantly from 0.017 h$^{-1}$ (MeSV2.1) to 0.081 ± 0.002 h$^{-1}$ (MeSV2.2), corresponding to an almost fivefold increase in doubling time (Fig. 4b, d). The population termed MeSV2.2 reached a maximum OD$_{600}$ that was 31 ± 4% higher than wild-type E. coli with empty plasmids (i.e., without the potential to grow on methanol) cultivated under the same conditions (Supplementary Fig. 3). Based on stoichiometric considerations, assuming no carbon loss, e.g., due to overflow metabolism, equimolar consumption of gluconate and methanol as shown in Fig. 2 would result in 17% higher maximal biomass compared to the sole consumption of a C6 substrate, indicating that the evolved MeSV2.2 surpassed equimolar methanol consumption. Comparative analysis of wild-type E. coli and MeSV2.2 supernatant with high-performance liquid chromatography suggested negligible carbon loss, since only a small accumulation of formic acid (0.09 ± 0.01 mM, in mid exponential phase) was detected.

### $^{13}$C-metabolic tracer analysis of evolved strains.

To characterize the evolved strains and to validate the incorporation of methanol into biomass, steady-state labeling experiments with $^{13}$C labeled methanol were performed (Fig. 5a). The evolved strains were cultivated in media containing natural labeled gluconate (5 mM) and $^{13}$C methanol (500 mM). Metabolome samples were collected after multiple doublings (>14 doublings) in exponential phase (OD$_{600}$ = 0.8) to ensure both metabolic and isotopic steady-state conditions. MeSV2 strains incorporated on average up to 23.9 ± 0.4% of methanol carbon into hexose 6-phosphate (Fig. 5b, c), the

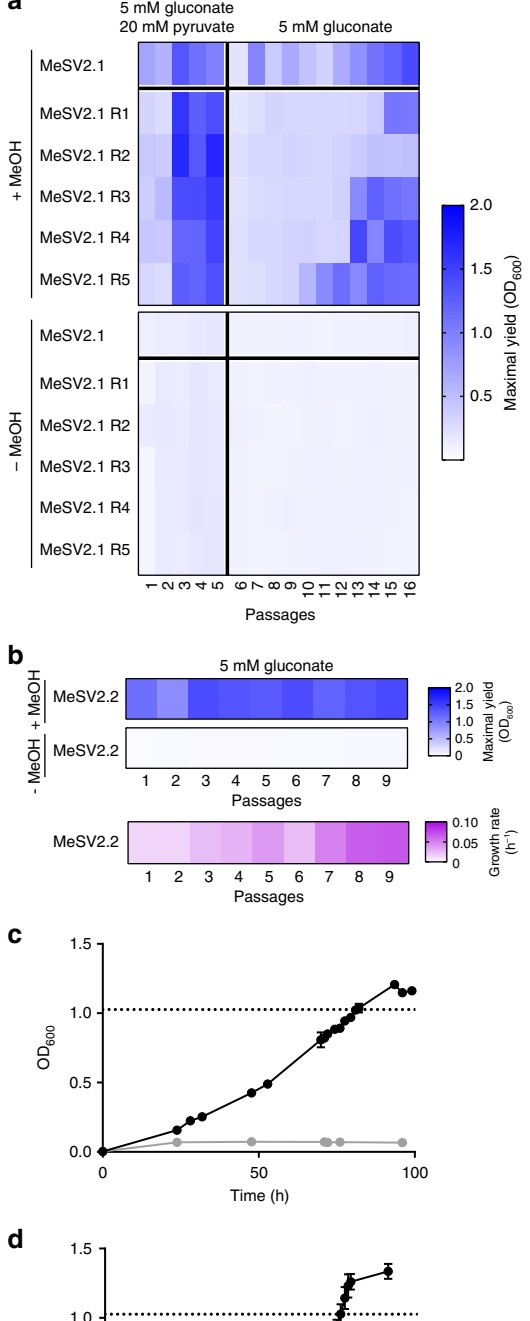

**Fig. 4** Evolution of methanol-essential strain version 2 (MeSV2). MeSV2 ($\Delta edd \Delta rpiA \Delta rpiB \Delta maldh$ pSEVA424 mdh2 PB1 pSEVA131 hps phi Mf) was evolved on 5 mM gluconate, and for the first five passages, with 20 mM pyruvate. Shown are maximal OD$_{600}$ values (blue) after each passage of MeSV2 with or without 500 mM methanol and with 0.1 g L$^{-1}$ yeast extract (**a**) or without yeast extract (**b**). The growth rate of cultures are shown in pink. Growth curves of the evolution endpoint with (black) and without (gray) methanol for strains MeSV2.1 on 5 mM gluconate with 0.1 g L$^{-1}$ yeast extract (**c**) and MeSV2.2 on 5 mM gluconate without yeast extract (**d**); n = 3. Data are presented as mean ± standard deviation. Dotted line represents final OD$_{600}$ of a wild-type E. coli strain containing empty plasmids under the same growth conditions

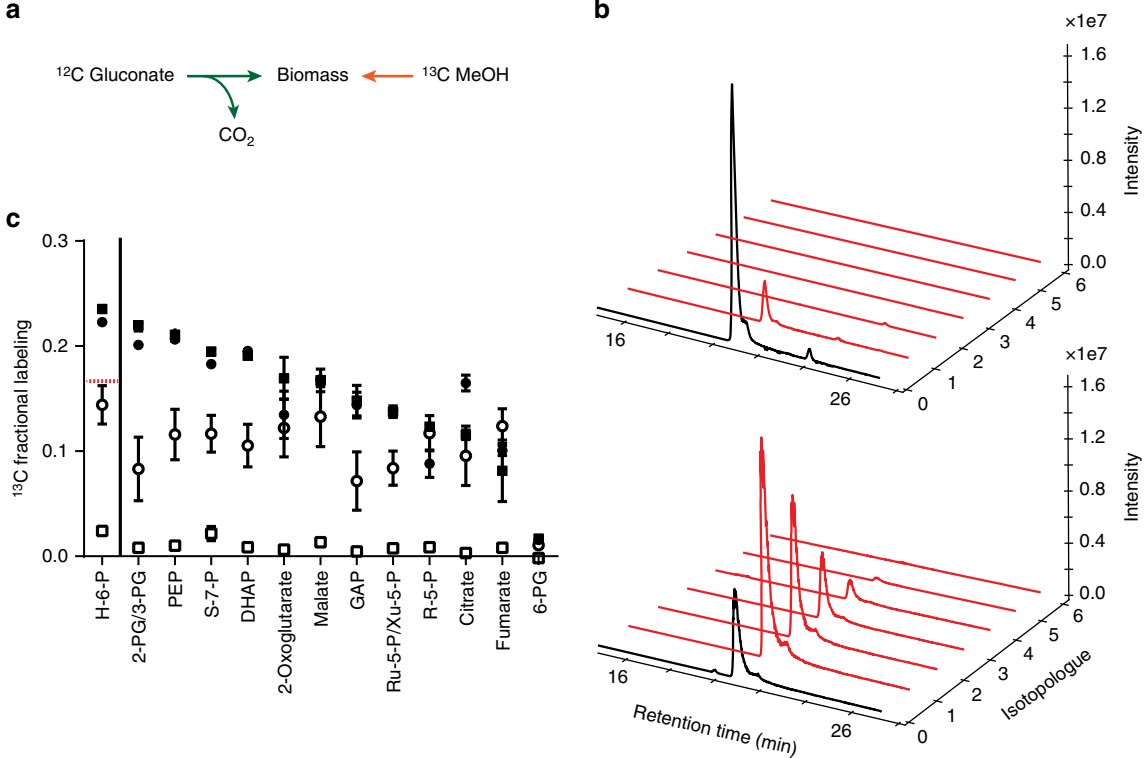

**Fig. 5** Incorporation of $^{13}C$ methanol during exponential growth. **a** Illustration of biomass formation from unlabeled gluconate and $^{13}C$ methanol. **b** Isotopologue distribution of hexose 6-phosphate in an *E. coli* wild-type strain containing *mdh* and *hps phi* (top) and MeSV2.2 (bottom). **c** Labeling detected in MeSV2.2 without yeast extract (closed squares) and MeSV2.1 grown with additional 0.1 g L$^{-1}$ unlabeled yeast extract (closed circles) are shown. For comparison, the results of *E. coli* wild type containing *mdh* and *hps phi* without yeast extract (open squares) and with 0.1 g L$^{-1}$ unlabeled yeast extract (open circles) are provided. Data presented as mean ± standard deviation; $n = 3$. The red dotted line represents the expected labeling fraction under equimolar consumption of gluconate and methanol

first multi-carbon compound produced from methanol assimilation (Fig. 2). This value was above the theoretical expected stoichiometric value of 17% and indicated that methanol consumption exceeded equimolar consumption of gluconate and is consistent with the yield increase mentioned above. In-line with the label incorporation, the M+1 isotopologue dominated (Fig. 5b). Other detected metabolites from glycolysis, the ribulose monophosphate cycle, and even TCA cycle metabolites showed an average carbon labeling in a similar range (11–24%; Fig. 5c). No major difference in labeling incorporation was observed for MeSV2.1 grown with yeast extract versus MeSV2.2 grown without yeast extract. This result indicates that the previously reported advantage in labeling incorporation due to yeast extract[28] is surpassed in the methanol-essential evolved strains.

In addition, we measured the methanol consumption rate of MeSV2.2 during exponential growth. To account for methanol evaporation and spontaneous methanol reactions, supernatants of control *E. coli* wild-type cultures containing empty plasmids were sampled in parallel. MeSV2.2 had a methanol consumption rate of $13 \pm 7$ mmol gCDW$^{-1}$ h$^{-1}$, which is in the same range as previously reported for natural methylotrophs (15 mmol g CDW$^{-1}$ h$^{-1}$)[44] and above recently published methanol co-metabolizing *E. coli* strains (0.019 gMeOH gCDW$^{-1}$ h$^{-1}$ corresponding to 0.6 mmol gCDW$^{-1}$ h$^{-1}$)[28].

**Genome sequencing of evolved strains**. Whole-genome sequencing was performed to identify the underlying genotypes responsible for the observed growth phenotype of the evolved strains. Overall, we identified 56 non-synonymous mutations in all evolution experiments combined. Five prominent gene mutations were detected multiple times with high frequencies (>20%) (Fig. 6), while most genes were only mutated once. Interestingly, two genes were mutated in all three laboratory evolution experiments and replicates (11/11). These genes encode the DNA-binding transcriptional repressor GntR (*gntR*) and the glutathione-dependent formaldehyde dehydrogenase (*frmA*). In the case of *gntR*, different missense mutations occurred, probably leading to an altered gluconate uptake. The predicted loss of function mutations observed in *frmA* (transposon insertions, deletions, or frameshift mutations) point toward a drain of formaldehyde via the direct oxidation pathway. This observation is in-line with the activation of the endogenous formaldehyde detoxification pathway in an Mdh expressing *E. coli* upon conversion of methanol to formaldehyde[26]. In three cases, we identified mutations in a phosphopentomutase (*deoB*), including two frame shifts, suggesting a loss of function. This enzyme catalyzes the transfer of a phosphate group between the C1 and C5 carbon atoms of ribose or deoxyribose, respectively, and represents a branch point of the RuMP-cycle to form 5-phosphoribosyl 1-pyrophosphate (PRPP)[45]. The suggested loss of function mutations in case of *frmA* and *deoB* is furthermore supported by in silico analysis, where KOs of these genes would in fact ensure flux distributions as predicted during growth on methanol as sole carbon source. Another mutation was observed in the gene encoding the transcription termination factor Rho (*rho*). Rho is required for mRNA transcription termination and acts as a global regulator of gene expression, which makes interpretation of this mutation difficult. Finally, we repeatedly observed mutations in the DNA-binding transcriptional repressor/nicotinamide

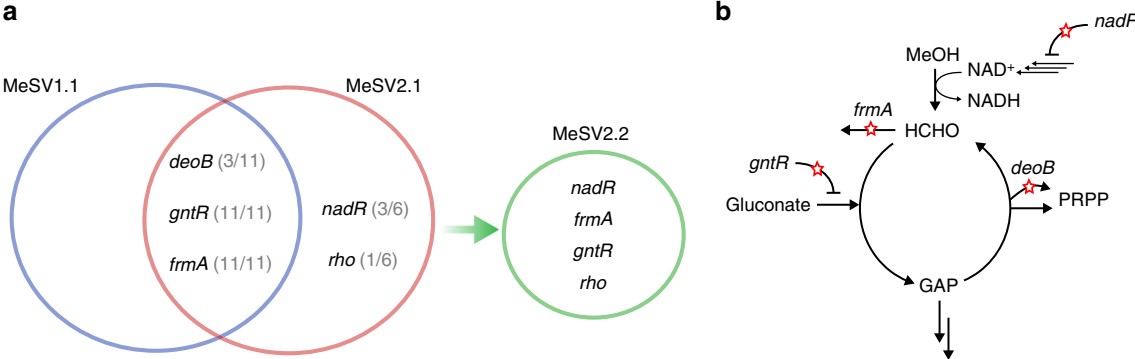

**Fig. 6** Prominent mutated genes in evolution experiments. **a** The fraction of mutations observed in independent evolution experiments is shown in brackets. Number of sequenced populations: MeSV1.1 $n = 5$, MeSV2.1 $n = 6$, MeSV2.2 $n = 1$. Mutations in *frmA* and *gntR* appear in all experiments whereas mutations in *nadR* and *rho* are only seen in MeSV2 strains. **b** Prominent mutated genes encoding enzymes in close proximity to branch points of the synthetic RuMP cycle are shown. For a detailed description of all mutations, see Supplementary Data 3

mononucleotide adenylyltransferase NadR (*nadR*). Mutations were exclusively found in the domain predicted to have ribosylnicotinamide kinase activity, which is involved in the NAD salvage pathway[46]. Accumulation of mutations in the N- and C-terminal domains have been described in previous long-term evolution experiments and are likely to reduce, rather than enhance, the function of the NAD repressor[47]. The mutations detected in *nadR* furthermore indicate the importance of the NAD$^+$/NADH ratio for methanol-dependent metabolism. No mutations were found in the heterologous genes *mdh*, *hps*, *phi*, and their promoters, indicating that the endogenous metabolism was limiting the accommodation of the introduced pathway rather than one-carbon conversion itself.

## Discussion

Engineering a cyclic carbon fixation pathway is a major challenge. To maintain cycle activity, a strictly balanced flux distribution, which allows for the recycling of the precursor responsible for the assimilation of the one-carbon compound, is required. This mode of operation implies an increased flux in the assimilation cycle and reduced loss of precursors via branch points to obtain a stable autocatalytic cycle[26, 32, 48]. In this study, we achieved methanol-essential growth of a non-methylotrophic organism. Using an in silico-guided KO approach, we first identified gene deletions predicted to result in methanol essentiality. By applying ALE experiments of engineered *E. coli* strains, we obtained evolved strains capable of methanol-dependent growth on gluconate. Coupling of methanol consumption to growth was crucial to obtain a selective condition needed for a successful evolution approach. In addition, prior knowledge regarding low TCA cycle activity in natural methylotrophs was important to escape a potential local fitness maximum and to obtain a strain capable of full growth rescue on gluconate. The effect of the NAD-dependent malate dehydrogenase KO supports the hypothesis that reduction in TCA cycle activity is metabolically beneficial for methylotrophy. However, the function of the NAD-dependent malate-dehydrogenase can be partially replaced by the malate:quinone oxidoreductase (*mqo*), or alternatively with a phosphoenolpyruvate (PEP) shunt via malic enzyme and PEP carboxylase. Nonetheless, it can be assumed that these alternative routes are only sustaining low TCA cycle activity. This hypothesis is strengthened by the finding that a Δ*maldh* strain grows with a reduced growth rate on pyruvate and lactate but fails to sustain growth on substrates where high TCA cycle activity are needed such as acetate or other C4 compounds[49].

The KO of NAD-dependent malate dehydrogenase might influence the NAD$^+$/NADH ratio in favor of the synthetic methanol metabolism. Under standard conditions, the NAD-dependent Mdh reaction is thermodynamically unfavored toward methanol oxidation $(\Delta_r G'^o = 34.2 \pm 6.5 \text{ kJ mol}^{-1})$[50]. One can assume that under reduced TCA cycle activity the NAD(H) ratio is re-balanced toward a high NAD$^+$/NADH ratio, which thermodynamically supports the oxidation of methanol. Moreover, due to decreased NADH production via the oxidative TCA cycle, the methanol dependency of the strain is increased. The importance of NAD$^+$/NADH ratios has been previously demonstrated, as it influences overflow metabolism and metabolic fluxes; additionally, it is considered an important factor for metabolic engineering[51]. All these considerations are supported by the mutation found in *nadR*.

Engineering methanol assimilation is, in many aspects, analogous to the challenge of implementing autotrophy. Interestingly, in a recently constructed hemi-autotrophic *E. coli* (CO$_2$ fixation during growth on pyruvate), ribose-phosphate diphosphokinase (*prs*) was identified as a mutated gene in three independent laboratory evolution experiments[32]. Further characterization indicated reduced flux from the one-carbon fixation cycle via PRPP into biomass. It is thus conceivable that the detected mutations in the phosphopentomutase (*deoB*) upon evolution to achieve methanol-essential growth also prevents draining of ribulose monophosphate cycle intermediates to PRPP biosynthesis.

The strains generated in this study will provide an ideal starting point for long-term chemostat evolution experiments where the multi-carbon source can be kept under limiting conditions while providing an excess of methanol. Using such a setup would allow methanol-dependent growth in the initial phase and could be used in a later stage to dynamically increase the selection stringency. As a more generally applicable strategy, the approach presented here might also be used to test alternative methanol conversion pathways beyond the ribulose-monophosphate pathway for one-carbon assimilation[52, 53]. In conclusion, the described concept and evolved strains represent a milestone in obtaining a truly synthetic methylotrophic organism and provide a successful example for optimizing a heterologous pathway by connecting it directly to cellular fitness.

## Methods

**Strain construction, media and reagents**. Gene KOs were constructed using P1 transduction using single KO strains from the KEIO collection as donor[54, 55]. After transduction, kanamycin-positive colonies were PCR-amplified to confirm gene deletions using primers up- and downstream of the introduced targeted deletion (Supplementary Table 1). For generating multiple KOs, the kanamycin resistance was removed by recombination using an FLP recombinase. The recombinase was introduced using the temperature sensitive pCP20 plasmid[56]. After recombinase

expression, loss of pCP20 and resistance cassette was confirmed by restreaking on the corresponding antibiotic and confirmation by polymerase chain reaction (PCR), followed by subsequent P1 transduction. All strains used in this study are listed in Supplementary Table 2.

[U$^{13}$C]-methanol (99%) was purchased from Cambridge Isotope Laboratories. All other chemicals, unless otherwise stated, were obtained from Sigma-Aldrich.

*E. coli* strains were cultivated in M9 minimal medium with (g L$^{-1}$): 6.78 Na$_2$HPO$_4$, 3 KH$_2$PO$_4$, 0.5 NaCl, 1 NH$_4$Cl, 0.049 MgSO$_4$·7H$_2$O, 0.0015 CaCl$_2$·2H$_2$O, 0.34 thiamine hydrochloride, trace elements (mg L$^{-1}$) 0.5 FeCl$_3$·6H$_2$O, 0.09 ZnSO$_4$·7H$_2$O, 0.088 CuSO$_4$·5H$_2$O, 0.045 MnCl$_2$, 0.09 CoCl$_2$·6H$_2$O with fitting antibiotics in the following concentrations (μg mL$^{-1}$): 100 ampicillin, 20 streptomycin sulfate, 50 kanamycin sulfate.

**Plasmid construction.** All plasmids used in this study were constructed by cloning the respective genes into pSEVA vectors[57] containing a Ptrc promotor (Supplementary Table 3). The genes were amplified from genomic DNA using the appropriate primers containing the ribosomal binding site AGGAGA upstream of the respective open reading frame (Supplementary Table 1). All constructs were verified by sequencing (Microsynth AG, Switzerland).

**In silico identification of methanol-essential genotypes.** KO analyses were performed using FlexFlux 1.0[37] based on the *E. coli* models iAF1260[36] and iML1515[39] containing additional reactions for NAD-dependent Mdh, as well as Hps and Phi. Constraints were defined by maximizing the objective function: R_Ec_biomass_iAF1260_core_59p81M or R_BIOMASS_Ec_iML1515_core_ 75p37M and setting the uptake rates for the different carbon sources to 7.0 mmol gCDW$^{-1}$ h$^{-1}$. Results for each multi-carbon compound were filtered for growth on methanol concurrent with a predicted growth rate of ≤0.01 h$^{-1}$ for the multi-carbon source alone. Differential fitness hits were obtained by comparing calculated growth rates on methanol versus co-consumption.

**ALE experiments.** Shake flask evolution experiments were performed in 20 mL M9 minimal medium containing 0.1 mM isopropyl-β-D-thiogalactopyranosid (IPTG) and appropriate antibiotics grown in 100 mL baffled shake flasks at 37 °C and 160 r.p.m. in a Minitron incubator (Infors HT). The medium was supplemented with a combination of the following carbon sources as noted throughout the study: 5 mM sodium gluconate, 20 mM sodium pyruvate, 0.1 g L$^{-1}$ yeast extract, and 500 mM methanol. A first passage was inoculated from an overnight culture grown at 37 °C, 160 r.p.m. in lysogeny broth (LB) medium containing 0.1 mM IPTG to induce gene expression. In general, inoculation of new passages was done after cultures reached stationary phase by centrifugation of 1 OD unit (1 mL of OD$_{600}$ = 1.0) at 3000×*g* for 5 min at room temperature, discarding of the supernatant, and re-suspension in the corresponding media. Cell growth was monitored at OD$_{600}$. Time required to reach stationary phase for evolution experiments MeSV1.1 and MeSV2.1 is shown in Supplementary Fig. 4. For every passage, a culture without methanol was grown as a control.

**$^{13}$C metabolic tracer analysis.** Cells were grown in the same manner as for the determination of maximal yield and growth rate but with $^{13}$C methanol. Metabolome samples were taken in three biological replicates during exponential growth[43]. Briefly, one OD unit of culture was sampled using a polyethersulfone (PESU) filter (0.2 μm, 47 mm diameter) and washed with 10 mL MilliQ water containing 1/10th of original carbon sources (2 mM unlabeled gluconate, 50 mM $^{13}$C methanol) kept at 37 °C. Immediately after washing, the filter was transferred to 8 mL pre-cooled (−20 °C) quenching solution (60:20:20 acetonitrile: methanol: 0.1 M formic acid) for extraction of intracellular metabolites. Subsequently, the samples were sonicated (Bath sonifier 2210, Branson), snap frozen, and lyophilized overnight. For liquid chromatography–mass spectrometry (LC–MS) analysis, the samples were re-suspended in 230 μM tributylamine (TBA), 3% methanol pH 9.0 to a final biomass concentration of 0.125 ng μL$^{-1}$ (assuming an OD unit per mgCDW correlation factor of 0.25).

$^{13}$C methanol incorporation into intracellular metabolites was analyzed by nanoscale high-performance liquid chromatography–high-resolution mass spectrometry (HPLC–HRMS) using an nLC-ultra system (Eksigent) hyphenated to a Q Exactive Plus (Thermo Scientific) instrument. Metabolite separation was achieved using a C18 column (Dr. Maisch Reprosil-Gold 120, 3 μm, 100 × 0.1 mm, Morvay Analytik) as the stationary phase and Solvent A/B as the mobile phase. Solvent A consisted of the ion-pairing reagent tributylamine (230 μM, TBA was first dissolved in 230 μM aqueous acetic acid; pH was adjusted to 9.0 with ammonium hydroxide). Methanol (Solvent B) was used as an eluent with the following multi-step gradient: 0 min, 0%; 35 min, 12%; 36 min, 90%; 48 min, 90%; 49 min, 0%; 60 min, 0%. The sample injection volume was 1 μL and the flow rate was 400 nL min$^{-1}$. Mass acquisition was operated in negative Fourier transform mass spectrometry (FTMS) in full MS scan mode. Obtained LC–MS labeling data were analyzed using eMZed2-based workflows[58]. Average labeling was calculated after targeted peak integration of different isotopes followed by correction for

natural labeling using Eq. (1):

$$\text{Labeled fraction} = \frac{\sum_{i=0}^{n} A_i \times i}{n \times \sum_{i=0}^{n} A_i}, \tag{1}$$

where $A_i$ is the abundance of the *i*th isotopologue and *n* the number of C atoms in the metabolite. Metabolites were identified by matching masses (mass tolerance of 3 milli mass units) and retention times with authenticated standards.

**Whole-genome sequencing.** Genomic DNA extraction was conducted according to the MasterPure$^{TM}$ DNA purification kit and sequencing was performed using an Illumina Hiseq 4000 platform at the Functional Genomics Center Zurich (FGCZ). The genome sequence of BW25113 GenBank: CP009273.1 including the plasmids encoding Mdh, Hps, and Phi were used as the reference for alignment and variant detection. Processing of obtained sequencing results was done using CLC Genomics Workbench Version 10 and Basic Variant Detection was carried out according to default settings.

**Methanol consumption.** Substrate conversion rates were calculated using Eq. (2) and data shown in Supplementary Fig. 5. The specific methanol uptake rate $q_{MeOH}$ was obtained according to Eq. (3). *t* represents the time in hours, $X_0$ the initial biomass concentration, and *μ* the specific growth rate in h$^{-1}$. Cellular dry weight (CDW) was calculated using a pre-defined correlation factor of 0.33 gCDW L$^{-1}$ OD$_{600}$[59].

$$\frac{dS_i(t)}{dt} = \frac{s_i(t_2) - s_i(t_1)}{t_2 - t_1}, \tag{2}$$

$$q_{MeOH} = \frac{dS_{MeOH}(t)/dt - dS_{evap}(t)/dt}{X_0 \times e^{\mu t}}. \tag{3}$$

Wild-type *E. coli* was used as an evaporation control. The specific growth rate *μ* was obtained using exponential regression on data obtained from three biological replicate measurements. Since all experimental measurement values have uncertainties, propagation of error was done according to the variance formula.

**Supernatant analysis.** Comparative supernatant analysis was performed based on supernatants collected after 8 h of exponential growth using an HPLC 1260 Inifinty II LC System (Agilent) equipped with a diode array detector (190–400 nm) and a Rezex ROA-Organic Acid H+ column (7.8 × 300 mm; Phenomenex) as analytical column. All samples were passed through a 0.20-μm filter prior to injection. Column temperature was kept at room temperature and mobile phase was sulfuric acid 5 mM with a flow rate of 0.55 mL min$^{-1}$ under isocratic condition.

**Data analysis.** Data are shown as mean ± standard deviation (*n* = 3), unless otherwise indicated. All evaluation experiments including growth curves, methanol consumption, and labeling experiments of evolved strains MeSV2.1 and MeSV2.2 have been carried out in three biological triplicates. ALE experiments have been carried out in replicate as follows: MeSV1.1 *n* = 5, MeSV2.1 *n* = 6, and MeSV2.2 *n* = 1.

**Data availability.** All data supporting the findings of this study are available in the article, Supplementary Information, or upon request from the corresponding author. The Illumina short reads generated in this study have been deposited at the European Nucleotide Archive (ENA) with the accession number PRJEB25370.

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

## Acknowledgements

We thank Philipp Christen for support with LC–MS measurements and Rémi Peyraud for support with FlexFlux. We are grateful to Ron Milo, Niv Antonovsky, Elad Noor, and Shmuel Gleizer for helpful discussion. This study was supported by a grant from the Swiss SystemsX.ch initiative within the framework of the ERA-Net ERASysAPP, MetApp, a grant from the Swiss National Science Foundation (31003A_173094), and ETH Zurich.

## Author contributions

F.M., P.K., and J.A.V. conceived the study. F.M. performed in silico modeling. F.M, Ph.K, J.H., and O.G.G. performed the experiments. F.M. supported by P.K. analyzed the data. F.M. and J.A.V. wrote the manuscript, with contributions from all authors.

## Additional information

**Competing interests:** The authors declare no competing interests.

