## [Peer Review File · Nature Communications]

Reviewers' comments:

Reviewer #1 (Remarks to the Author):

In their manuscript, Meyer et al. develop a strain of *Escherichia coli* and associated conditions for which the growth of the strain is dependent on the presence of methanol. Their strategy involved the use of a cosubstrate, gluconate, to supply the five carbon intermediates of RuMP, while knocking out the genes that allow *E. coli* to grow on gluconate alone. This, along with their insight that maintaining a high NAD⁺/NADH ratio favors methanol oxidation, which they accomplished by reducing TCA cycle activity, resulted in a strain that grew quite readily in gluconate and methanol without complex carbon sources such as yeast extract.

The reported work is a well-designed and well-reasoned combination of rational and non-rational approaches. The insight is novel and differs from that of previously reported approaches. The development of methanol dependency in *E. coli* could be a potential stepping stone towards synthetic methylotrophy, as the authors suggest. However, I feel the authors should address the following issues before the manuscript is ready for publication:

- Why were there not more replicates conducted and sequenced for the MeSV2.2 evolution experiments as was done with MeSV2.1? It seems like this is the most interesting result with the highest methanol consumption rate and reaching the highest ODs and so should have the greatest amount of analysis. From my understanding, the MeSV2.2 experiments were initiated using the MeSV2.1 population, so it would also be worthwhile to delve deeper into the differences between the two populations. For example, it is interesting that, looking at the data, MeSV2.1 has a Gly99Val mutation in rho, but in MeSV2.2, that location has reverted to the wild type and instead there is a Arg299His mutation. Does this happen consistently and is it required to achieve this phenotype?
- Fig. 6 is misleading if the MeSV2.2 experiment was performed as initiated with the MeSV2.1 population. As shown, it seems that MeSV2.2 and MeSV2.1 are independent experiments that separately resulted in, for example, nadR mutations. It would seem, though, that the mutation in nadR resulting in MeSV2.2 is simply passed from MeSV2.1. Had MeSV2.1R1 been used to initiate the MeSV2.2 experiment, would the nadR mutation be present?
- It would be very beneficial to include at least time between passages (i.e. time required to reach the OD of each passage) or better the growth rate (as in Fig. 4b) for all evolution experiments. The OD alone may not be indicative of the fitness of the population especially if the time between passages varies. This is demonstrated by Fig. 4b, where final OD is relatively constant

but growth rate increases. If the time between passages is constant, this should be stated. If not, then the method for determination of when to passage should be mentioned.

- In Fig. 4C, why does it appear that MeSV2.1 has a much shorter lag phase than MeSV2.2? Was yeast extract present in the experiments with MeSV2.1? If so, it may not be appropriate to compare the two growth curves in the same chart. At least, this should be clearly mentioned, but perhaps it would be better split this into two plots.

- In Fig. 6, the caption reads that “mutations in nadR, deoB, and rho are only seen in MeSV2 strains”, but the diagram shows deoB as occurring in MeSV1 strains.

- In Fig. 2, the effect of the RpiAB knockout might be clearer if a bidirectional arrow were used for the reaction, indicating that flux could lead away from Ru5P with RpiAB present.

- The comparison of growth in Fig. 3b would benefit from more replicates and error bars to demonstrate that the observed additional growth is significant.

- In the second sentence of the abstract, the statement that “Utilization of one-carbon compounds for growth is limited to methylotrophic organisms” would be better restricted to “reduced one-carbon compounds”.

Reviewer #2 (Remarks to the Author):

The manuscript “Methanol-essential growth of Escherichia coli” submitted for publication in Nature Communications by Meyer et al. describes the design and construction of an E. coli strain that requires methanol for growth. Laboratory evolution was utilized to further increase methanol consumption. Gene mutations of the evolved strain were elucidated to gain insight into metabolism re-routing.

The study describes the first methanol-essential non-methylotrophic organism and lays the foundation for future synthetic biology approaches in the quest for efficient conversion of C1 compounds.

GENERAL COMMENTS

This is a very timely topic of high interest to metabolic engineering and sustainability research. The manuscript is well written, balanced and easy to follow. There are a few points that should be addressed by the authors (see specific comments below). In particular it would be interesting (and needed for balancing) to check the supernatant for any product formation in the mutant strains. The engineering of a methanol-essential strain is a great achievement in itself, but I would argue that this strain should be able to perform as good (or better) than co-metabolizing strains. The authors should thus compare their data with existing strains, e.g. a recent publication by Gonzalez et al. (Met Eng 45: 67-74, 2018). It would be crucial to evaluate the differences in methanol conversion by the methanol-essential strains to co-metabolizing strains developed in this study. The authors should comment hereby also on conversion rates (not only yield) and

highlight why their strain would represent a superior route over co-metabolism. This would strengthen the manuscript even further and would justify bold claims such as “represent a milestone in obtaining a truly synthetic methylotrophic organism”. In addition, I would recommend that the authors rerun some of their analysis using updated models to include state-of-the-art knowledge, which would potentially lead to new insights and more comprehensive analysis given the significant larger scope of these models.

SPECIFIC COMMENTS

L24 Please rephrase – the fact that CH₄ is a more potent greenhouse gas than CO₂ does not make it a key player in the carbon cycle...

L64 While the model (iAF1260) by Feist et al from 2007 produced sufficient results, it would be important to run the workflow again with an up to date model (iJO1366 or iML1515) and report and update the results shown in Supplementary data 1.

L82 N.b. Nota bene? Can be deleted.

L85 KOs instead of knockouts

L109-110 Why were genes from these two organisms chosen? Have other genes be tested?

L112 Fig. S2a – there is hardly any growth increase in this figure. Without this experiment being performed in triplicate (no error bars in this figure?), it is hard to tell if there is actually any effect of methanol on the growth phenotype of this strain. Also, better to show a y-axis with a lower max (0.4?). I assume they want to be consistent throughout the manuscript but it's hard to tell anything from a growth curve with this resolution.

L117 A much more rigorous analysis would be needed for this statement. What regulation are the authors referring to?

L122 Fig.3 again, why showing growth curves with y-axis of 1.5 when the final OD barely reaches 0.3? The growth of the WT strain can be described in the legend as final OD 1.0 (the dotted line is not providing any further information to that).

Please provide error bars in the growth figure so that the reader can evaluate replicate differences.

L148 The growth is quite peculiar. There is a very long lag phase for MeSV2.2. Why is this? Has this been observed all the time? Is this due to inoculum size and thus maybe toxicity? MeSV2.1 on the other hand hardly exhibits exponential growth. What is the explanation for this? While the authors focus on the final OD, it would be important to show the growth of a WT E. coli in this figure as well to highlight phenotypic behavior of WT vs. engineered strains.

L212 Maybe I am missing this but running the model with the gene mutations (e.g. loss of function of frmA and others) would be crucial and would contribute additional information who fluxes changes based on the mutations. It would also help to integrate the mutation analysis into the strain design part of the manuscript. As it is, this part is very descriptive and “stands alone”.

L235 This is probably beyond the scope of this study, but did the authors perform supplementation experiments to further increase yield or rate increase using the final mutant

strain and to explore favorable NAD/NADH ratios (see L236ff)?

L252 It would be very interesting to know if the strains produce any products (in the supernatant). Simple metabolomics would be able to identify any overflow metabolism and should be included in the study.

Fig. S1 Clock-wise orientation of the figure a-c would be preferable.

Supplementary Data 1 if not formatted correctly (cut-off text appears on page 2)

Reviewer #3 (Remarks to the Author):

The manuscript by Meyer et al. describes the development of an E.coli strain that can co-utilize methanol and gluconate and can't grow in the absence of methanol. This has been achieved by reducing the flux through the TCA cycle and by multiple rounds of Adaptive Laboratory Evolution. Important mutations have been identified through whole genome sequence.

The manuscript presents novel information. I found particularly insightful the work that was done around the TCA cycle. However there are a number of serious issues, some of which relate to the way the manuscript has been written.

To start with, the initial reading of the introduction gave the impression that the authors have developed an E.coli strain that is able to grow exclusively on methanol. Clearly this is not the case and this is something that needs to be emphasized in the last paragraph of the introduction.

Line 71: How did the authors come up with number 7 mmol /gCDW/hr used for the uptake rate of the multi-carbon source? Did they test different uptake rates?

Line 89: it appears that the identified multiple knock-out targets was based on stacking additional knock-outs on previously identified single knockouts. My impression from reading this section is that the authors did not attempt combinatorial approaches to identify multiple simultaneous knock-outs. This could be a potential problem given the fact that often times stacking of additional knockouts on previously identified single knockouts does not result in the identification of a global optimum. This has been demonstrated in doi: 10.1128/AEM.00270-09 and doi: 10.1016/j.ymben.2011.06.008

Line 109: how was *Bacillus methanolicus* methanol dehydrogenase selected?

Line 155: I am confused about the yeast extract. The previous ALE experiments that resulted in

the generation of strain MeSV2.1 were performed in the presence of pyruvate. My understanding is that the OD of 1.21 achieved by this strain is the presence of gluconate and methanol only. Therefore, I don't understand what is the point of further optimizing MeSV2.1 in the "absence of yeast extract". Going through the experimental methods section, it appears that yeast extract might be present throughout the evolutionary experiments, however this is not clear when reading the Results section.

Genome sequence results: it would be interesting to know if the authors have attempted reverse metabolic engineering with the identified mutants. In other words, what is the minimum number of mutations that will result in a phenotype that allows growth on gluconate with the required presence of methanol?

Responses to Reviewers:

Reviewers' comments:

Reviewer #1 (Remarks to the Author):

In their manuscript, Meyer et al. develop a strain of *Escherichia coli* and associated conditions for which the growth of the strain is dependent on the presence of methanol. Their strategy involved the use of a cosubstrate, gluconate, to supply the five carbon intermediates of RuMP, while knocking out the genes that allow *E. coli* to grow on gluconate alone. This, along with their insight that maintaining a high NAD⁺/NADH ratio favors methanol oxidation, which they accomplished by reducing TCA cycle activity, resulted in a strain that grew quite readily in gluconate and methanol without complex carbon sources such as yeast extract.

The reported work is a well-designed and well-reasoned combination of rational and non-rational approaches. The insight is novel and differs from that of previously reported approaches. The development of methanol dependency in *E. coli* could be a potential stepping stone towards synthetic methylotrophy, as the authors suggest. However, I feel the authors should address the following issues before the manuscript is ready for publication:

- Why were there not more replicates conducted and sequenced for the MeSV2.2 evolution experiments as was done with MeSV2.1? It seems like this is the most interesting result with the highest methanol consumption rate and reaching the highest ODs and so should have the greatest amount of analysis. From my understanding, the MeSV2.2 experiments were initiated using the MeSV2.1 population, so it would also be worthwhile to delve deeper into the differences between the two populations. For example, it is interesting that, looking at the data, MeSV2.1 has a Gly99Val mutation in rho, but in MeSV2.2, that location has reverted to the wild type and instead there is a Arg299His mutation. Does this happen consistently and is it required to achieve this phenotype?

The main aim of this study was to create a methanol-essential strain, which was achieved in case of the evolution experiments conducted in multiple replicates and resulting in MeSV2.1. The difference between MeSV2.1 and MeSV2.2 is the improved growth without small amounts of yeast extract (0.01%). The fact that critical mutations required for increased methanol conversion were present already in MeSV2.1 is evident also from the labeling data shown in Fig. 5c (closed symbols). Regarding the difference between MeSV2.1 and MeSV2.2, population sequencing of MeSV2.1 showed that the rho mutation Gly99Val had a frequency of 52% in the population whereas in case of MeSV2.2 the frequency of the Arg299His mutation was only present in 23% of the population, suggesting that "reversion" to wild type is a predominant feature during transition from MeSV2.1 to MeSV2.2. This information was previously not available to the reader and we updated the supplementary Data 1. Furthermore, we noticed that the excel sheet provided with the initial submission was incomplete and lacked some minor mutations at low frequency that are not discussed in the main text. This is now corrected.

- Fig. 6 is misleading if the MeSV2.2 experiment was performed as initiated with the MeSV2.1 population. As shown, it seems that MeSV2.2 and MeSV2.1 are independent experiments that separately resulted in, for example, nadR mutations. It would seem, though, that the mutation in nadR resulting in MeSV2.2 is simply passed from MeSV2.1. Had MeSV2.1R1 been used to initiate the MeSV2.2 experiment, would the nadR mutation be present?

We agree and adapted Figure 6 to clarify that MeSV2.2 was initiated from MeSV2.1. Moreover, the figure was complemented with a scheme to visualize the mutated genes/gene products with respect to their metabolic function. Regarding the question on the evolution of

MeSV2.1R1, we did not initiate an evolution experiment from MeSV2.1R1. However, the increase in population frequency of the mutation in *nadR* from MeSV2.1 (68%) to MeSV2.2 (98%) indeed suggests a relevance of the mutation also in the case of MeSV2.2.

- It would be very beneficial to include at least time between passages (i.e. time required to reach the OD of each passage) or better the growth rate (as in Fig. 4b) for all evolution experiments. The OD alone may not be indicative of the fitness of the population especially if the time between passages varies. This is demonstrated by Fig. 4b, where final OD is relatively constant but growth rate increases. If the time between passages is constant, this should be stated. If not, then the method for determination of when to passage should be mentioned.

Indeed, the time between passages varied. This is now clarified in the material and methods section. Likely, due to the presence of yeast extract for MeSV1 and MeSV2.1, no exponential growth was observed and the calculation of a physiological growth rate is not possible. To address the concern of the reviewer, time required to reach final OD for the evolution experiments with the strains MeSV1 and MeSV2.1 is now shown in two separate figures (Supplementary Fig. S4).

- In Fig. 4C, why does it appear that MeSV2.1 has a much shorter lag phase than MeSV2.2? Was yeast extract present in the experiments with MeSV2.1? If so, it may not be appropriate to compare the two growth curves in the same chart. At least, this should be clearly mentioned, but perhaps it would be better split this into two plots.

Yeast extract was present in case of MeSV2.1, but not in MeSV2.2. We followed the recommendation of the reviewer and splitted the figure in two separate plots and added the information on the presence/absence of yeast extract to the Figure legend. The lag phase of MeSV2.2 is reproducible under the given conditions. We speculate the difference is due to the known phenomenon caused by too low concentrations of CO₂, which are required to fuel anaplerotic CO₂-dependent enzymes and occurs when small inoculum sizes are used (Repaske & Clayton, 1978), whereas in case of MeSV2.1 yeast extract can compensate this effect due to the lower demand in anaplerotic reactions.

- In Fig. 6, the caption reads that “mutations in *nadR*, *deoB*, and *rho* are only seen in MeSV2 strains”, but the diagram shows *deoB* as occurring in MeSV1 strains.

The caption was corrected accordingly.

- In Fig. 2, the effect of the RpiAB knockout might be clearer if a bidirectional arrow were used for the reaction, indicating that flux could lead away from Ru5P with RpiAB present. Figure 2 was modified accordingly.

- The comparison of growth in Fig. 3b would benefit from more replicates and error bars to demonstrate that the observed additional growth is significant.

Growth shown in Fig 3b has been repeated in triplicates and is shown with error bars (they are for most data points below ± 0.008). The y-axis has been changed according to the reviewers comment.

- In the second sentence of the abstract, the statement that “Utilization of one-carbon compounds for growth is limited to methylotrophic organisms” would be better restricted to “reduced one-carbon compounds”.

The sentence was changed as suggested.

Reviewer #2 (Remarks to the Author):

The manuscript “Methanol-essential growth of *Escherichia coli*” submitted for publication in Nature Communications by Meyer et al. describes the design and construction of an *E. coli* strain that requires methanol for growth. Laboratory evolution was utilized to further increase methanol consumption. Gene mutations of the evolved strain were elucidated to gain insight into metabolism re-routing.

The study describes the first methanol-essential non-methylotrophic organism and lays the foundation for future synthetic biology approaches in the quest for efficient conversion of C1 compounds.

GENERAL COMMENTS

This is a very timely topic of high interest to metabolic engineering and sustainability research. The manuscript is well written, balanced and easy to follow. There are a few points that should be addressed by the authors (see specific comments below). In particular it would be interesting (and needed for balancing) to check the supernatant for any product formation in the mutant strains. The engineering of a methanol-essential strain is a great achievement in itself, but I would argue that this strain should be able to perform as good (or better) than co-metabolizing strains. The authors should thus compare their data with existing strains, e.g. a recent publication by Gonzalez et al. (Met Eng 45: 67-74, 2018).

It would be crucial to evaluate the differences in methanol conversion by the methanol-essential strains to co-metabolizing strains developed in this study. The authors should comment hereby also on conversion rates (not only yield) and highlight why their strain would represent a superior route over co-metabolism. This would strengthen the manuscript even further and would justify bold claims such as “represent a milestone in obtaining a truly synthetic methylotrophic organism”.

Comparison of these two conditions is quite difficult. In the mentioned paper, the authors used a complex media (yeast extract, 1.5 g/L) and methanol consumption occurs predominantly in the stationary phase. In our study, methanol is required for growth and consumed at all times. To address the reviewers comment we compared the uptake rate with the recent publication of Whitaker et al. ($0.019 \text{ gMeOH gCDW}^{-1} \text{ h}^{-1}$ corresponding to $0.6 \text{ mmol gCDW}^{-1} \text{ h}^{-1}$) to the one we achieved in our study ($13 \text{ mmol gCDW}^{-1} \text{ h}^{-1}$) and added the comparison to the manuscript.

In addition, I would recommend that the authors rerun some of their analysis using updated models to include state-of-the-art knowledge, which would potentially lead to new insights and more comprehensive analysis given the significant larger scope of these models.

Rerunning our analysis with an updated model (iML1515) showed that the identified methanol essentiality on gluconate is confirmed using the genotype $\Delta\text{edd}\Delta\text{rpiAB}$. As some other predictions between both models deviate (see below), we additionally added the results of this analysis to the Supplemental material (Supplementary Data 2) and adapted the results section.

SPECIFIC COMMENTS

L24 Please rephrase – the fact that CH₄ is a more potent greenhouse gas than CO₂ does not make it a key player in the carbon cycle...

The sentence was changed accordingly.

L64 While the model (iAF1260) by Feist et al from 2007 produced sufficient results, it would

be important to run the workflow again with an up to date model (iJO1366 or iML1515) and report and update the results shown in Supplementary data 1.

Rerunning our analysis with the most recent model published in 2017 (iML1515) showed that the identified methanol essentiality on gluconate is confirmed using the genotype $\Delta\text{edd}\Delta\text{rpiAB}$. As mentioned above, we re-analysed the entire workflow with the model iML1515 and the results are now added to Supplementary Data 2. In a few cases, we identified new methanol-essential genotypes whereas most identified methanol-essential genotypes remained the same. Furthermore, some previously identified methanol-essential genotypes in case of iAF1260 were not identified using iML1515.

L82 N.b. Nota bene? Can be deleted.

Deleted.

L85 KOs instead of knockouts

Changed.

L109-110 Why were genes from these two organisms chosen? Have other genes be tested?

In a previous study, we compared *in vivo* activities of various enzymes including Mdh2 from *B. methanolicus* PB1 and HpsPhi from *M. flagellatus* (Müller *et al.*, 2015). To address the reviewer's concern we now refer to the earlier publication in which the reader can find further details.

L112 Fig. S2a – there is hardly any growth increase in this figure. Without this experiment being performed in triplicate (no error bars in this figure?), it is hard to tell if there is actually any effect of methanol on the growth phenotype of this strain. Also, better to show a y-axis with a lower max (0.4?). I assume they want to be consistent throughout the manuscript but it's hard to tell anything from a growth curve with this resolution.

The experiment for Fig. S2 has been repeated in triplicates and the y-axis has been adapted according to the reviewer's suggestion. In case of Fig. S2a only a slight growth advantage due addition of methanol was observed whereas with gluconate and pyruvate (Fig 2b) this difference is drastically increased.

L117 A much more rigorous analysis would be needed for this statement. What regulation are the authors referring to?

According to FBA, growth on pyruvate should be possible without methanol whereas growth on gluconate without methanol is not possible. Because no growth on gluconate in combination of pyruvate was observed, we suppose that metabolic regulation is responsible for that phenotype. We modified the sentence to improve clarity.

L122 Fig.3 again, why showing growth curves with y-axis of 1.5 when the final OD barely reaches 0.3? The growth of the WT strain can be described in the legend as final OD 1.0 (the dotted line is not providing any further information to that).

Please provide error bars in the growth figure so that the reader can evaluate replicate differences.

The experiment has been repeated in triplicates and the y-axis has been adapted according to the suggestions. We now included the information on the final OD₆₀₀ of the wild type strain under this condition in the caption.

L148 The growth is quite peculiar. There is a very long lag phase for MeSV2.2. Why is this? Has this been observed all the time? Is this due to inoculum size and thus maybe toxicity?

The lag phase might be due to the inoculum size and shows up repeatedly. This is likely due to a known phenomenon caused by too low metabolic CO₂ concentrations needed to fuel different CO₂-dependent pathways if a small inoculum size is used (Repaske & Clayton, 1978).

MeSV2.1 on the other hand hardly exhibits exponential growth. What is the explanation for this?

MeSV2.1 is growing with supplemented yeast extract, therefore providing an initial complex nutrient source that is changing over time. The peculiar growth of MeSV2.1 might arise from metabolic adaptations during the different growth phases.

While the authors focus on the final OD, it would be important to show the growth of a WT *E. coli* in this figure as well to highlight phenotypic behavior of WT vs. engineered strains. Growth curves of WT *E. coli* versus MeSV2.2 under the same growth conditions are now shown in Supplementary Figure 4. Whereas wild type *E. coli* is able to grow fast in a methanol independent way on gluconate (as expected), our evolved strain grows slower but reaches higher final OD due to methanol utilization.

L212 Maybe I am missing this but running the model with the gene mutations (e.g. loss of function of *frmA* and others) would be crucial and would contribute additional information who fluxes changes based on the mutations. It would also help to integrate the mutation analysis into the strain design part of the manuscript. As it is, this part is very descriptive and “stands alone”.

We included mutations in genes encoding metabolic enzymes into the model to check for their contribution. In case of *deoB* and *frmA*, we are assuming a loss of function due to frameshift mutations and transposon insertions. *In silico* analysis of the proposed loss of function mutations of in these genes ensures flux distribution as predicted by methanol as sole carbon source. This result is now included in the manuscript.

L235 This is probably beyond the scope of this study, but did the authors perform supplementation experiments to further increase yield or rate increase using the final mutant strain and to explore favorable NAD/NADH ratios (see L236ff)?

Additional experiments to explore NAD⁺/NADH ratios might indeed increase growth rate and yield of the evolved strain even further and could be used as a next step. However, so far we did not conduct such experiments.

L252 It would be very interesting to know if the strains produce any products (in the supernatant). Simple metabolomics would be able to identify any overflow metabolism and should be included in the study.

Using HPLC experiments (ROA-Organic Acid H⁺ (8%)) and complementary NMR experiments, we investigated culture supernatants of MeSV2.2 growing on 5 mM gluconate and 500 mM methanol. The sole detectable product was formic acid, which accumulated up to 0.09 ± 0.01 mM after 8 hours of exponential growth. Due to the loss of function mutation of *frmA*, other reactions must be responsible for minor formic acid production, e.g. catalyzed by pyruvate formate-lyase. To address the reviewers comment we added the information that no major products were detectable to the result section of the manuscript.

Fig. S1 Clock-wise orientation of the figure a-c would be preferable. The Supplementary Figure S1 was adjusted accordingly.

Supplementary Data 1 if not formatted correctly (cut-off text appears on page 2)

The issue has been resolved.

Reviewer #3 (Remarks to the Author):

The manuscript by Meyer et al. describes the development of an E.coli strain that can co-utilize methanol and gluconate and can't grow in the absence of methanol. This has been achieved by reducing the flux through the TCA cycle and by multiple rounds of Adaptive Laboratory Evolution. Important mutations have been identified through whole genome sequence.

The manuscript presents novel information. I found particularly insightful the work that was done around the TCA cycle. However there are a number of serious issues, some of which relate to the way the manuscript has been written.

To start with, the initial reading of the introduction gave the impression that the authors have developed an E.coli strain that is able to grow exclusively on methanol. Clearly this is not the case and this is something that needs to be emphasized in the last paragraph of the introduction.

To avoid misunderstandings, we modified the introduction accordingly and put more emphasis on the requirement for a co-substrate.

Line 71: How did the authors come up with number 7 mmol /gCDW/hr used for the uptake rate of the multi-carbon source? Did they test different uptake rates?

The 7 mmol/gCDW/h were chosen because it lies in the range of uptake rates reported under shake flask conditions (e.g. glucose (Fischer *et al*, 2004)). This information was added to the manuscript. Altering uptake rates had no effect on predicted methanol essential genotypes.

Line 89: it appears that the identified multiple knock-out targets was based on stacking additional knock-outs on previously identified single knockouts. My impression from reading this section is that the authors did not attempt combinatorial approaches to identify multiple simultaneous knock-outs. This could be a potential problem given the fact that often times stacking of additional knockouts on previously identified single knockouts does not result in the identification of a global optimum. This has been demonstrated in doi:

10.1128/AEM.00270-09 and doi: 10.1016/j.ymben.2011.06.008

We agree that a combinatorial approach could also be used and we agree that the applied computational setup might potentially miss some methanol-essential genotypes. However, a combinatorial approach using the entire model with its more than 2000 reactions would exceed a feasible calculation time - even when only considering double knockouts. To minimize this calculation time one could test a combinatorial approach on a subset of "important reactions". In this case, we ultimately would lose the unbiased character of our analysis due to the preselection of a subnetwork only.

Line 109: how was *Bacillus methanolicus* methanol dehydrogenase selected

In a previous study, we compared *in vivo* activities of various enzymes including Mdh2 from *B. methanolicus* PB1 and HpsPhi from *M. flagellatus* (Müller *et al*, 2015). To address the reviewer's concern we now refer to this publication when we mention the enzymes the first time.

Line 155: I am confused about the yeast extract. The previous ALE experiments that resulted in the generation of strain MeSV2.1 were performed in the presence of pyruvate. My understanding is that the OD of 1.21 achieved by this strain is the presence of gluconate and methanol only. Therefore, I don't understand what is the point of further optimizing MeSV2.1 in the "absence of yeast extract". Going through the experimental methods section, it appears that yeast extract might be present throughout the evolutionary experiments, however this is not clear when reading the Results section.

As correctly stated by the reviewer in all ALE experiments except MeSV2.2, 0.1 g/L yeast extract was present in addition to gluconate and methanol as mentioned in the material and methods part. For clarification, we adapted the text of the results section to include this information.

Genome sequence results: it would be interesting to know if the authors have attempted reverse metabolic engineering with the identified mutants. In other words, what is the minimum number of mutations that will result in a phenotype that allows growth on gluconate with the required presence of methanol?

So far, reverse metabolic engineering with the identified mutations has not been done and was beyond the scope of this study. We used *in silico* analysis of the proposed loss of function mutations of the two genes *deoB* and *frmA*, as suggested by reviewer 2. These showed that knockouts of these genes would in fact ensure flux distributions as predicted by methanol as sole carbon source.

Fischer E, Zamboni N & Sauer U (2004) High-throughput metabolic flux analysis based on gas chromatography-mass spectrometry derived ^{13}C constraints. *Anal. Biochem.* **325**: 308–316

Müller JEN, Meyer F, Litsanov B, Kiefer P, Potthoff E, Heux S, Quax WJ, Wendisch VF, Brautaset T, Portais JC & Vorholt JA (2015) Engineering *Escherichia coli* for methanol conversion. *Metab. Eng.* **28**: 190–201 Available at: <http://dx.doi.org/10.1016/j.ymben.2014.12.008>

Repaske R & Clayton MA (1978) Control of *Escherichia coli* growth by CO₂. *J. Bacteriol.* **135**: 1162–1164

Reviewers' comments:

Reviewer #1 (Remarks to the Author):

The authors have satisfactorily addressed all my comments/suggestions.

Reviewer #2 (Remarks to the Author):

The authors now state that the very long lag phase of strain MeSV2.2 can be attributed to not sufficient CO₂ concentrations during initial growth phase. Confirming this by increasing CO₂ in the atmosphere would be worthwhile to show. Beside this, the authors have addressed all my concerns and comments. The revised manuscript has improved substantially and the experiments and results are clearly presented.

Reviewer #3 (Remarks to the Author):

Overall the response to the reviewers comments was satisfactory. I would just like to point out that there is a number of different computational approaches that have been developed to address the issue of combinatorial explosion in FBA studies with multiple mutants. These algorithms specifically address the problem of computation power necessary that stems from the huge number of phenotypes that need to be evaluated when exploring different numbers of gene knockouts.

Responses to Reviewers:

Reviewer #1 (Remarks to the Author):

The authors have satisfactorily addressed all my comments/suggestions.

Reviewer #2 (Remarks to the Author):

"The authors now state that the very long lag phase of strain MeSV2.2 can be attributed to not sufficient CO₂ concentrations during initial growth phase. Confirming this by increasing CO₂ in the atmosphere would be worthwhile to show. Beside this, the authors have addressed all my concerns and comments. The revised manuscript has improved substantially and the experiments and results are clearly presented."

In our previous response, we speculated about a potential CO₂ effect. After careful investigation of the growth phenotype of MeSV2.2 we concluded that although it appeared that the strain exhibits an initial lag phase, the difference to a simulated exponential growth curve (without lag phase) is almost not distinguishable. See graph below (overlay of simulated red and measured values black). We apologize; we should have pointed this out in our previous response. Moreover, the variance between measured and simulated growth is close to the technical limit of detection (measured values below 0.015) at initial time point. The minor difference between these growth curves might indeed be caused by a low initial CO₂ concentration. However, we think that it is rather not worthwhile repeating the assay. The main point of the data shown in Figure 4 is to illustrate the increased yield due to methanol consumption compared to a wild type strain (dotted line).

Reviewer #3 (Remarks to the Author):

Overall the response to the reviewers comments was satisfactory. I would just like to point out that there is a number of different computational approaches that have been developed to address the issue of combinatorial explosion in FBA studies with multiple mutants. These algorithms specifically address the problem of computation power necessary that stems from the huge number of phenotypes that need to be evaluated when exploring different numbers of gene knockouts.

We would like to thank the reviewer for pointing out these approaches. In fact, we agree that combinatorial explosion in FBA studies can be addressed using these algorithms. Nonetheless, we would like to highlight that with our approach feasible solutions have been found and were confirmed experimentally.

Reviewers' comments:

Reviewer #2 (Remarks to the Author):

The figure explains the growth effect well.
The manuscript should be published as is now.

Reviewer #3 (Remarks to the Author):

nothing else to add.